

# Comparing lightning observations of the ground-based EUCLID network and the space-based ISS-LIS

Dieter R. Poelman[1], Wolfgang Schulz[2]

[1]Royal Meteorological Institute of Belgium, Brussels, Belgium
[2]OVE-ALDIS, Vienna, Austria

*Correspondence to*: Dieter R. Poelman (dieter.poelman@meteo.be)

**Abstract.** The Lightning Imaging Sensor (LIS) on the International Space Station (ISS) detects lightning from space by capturing the optical scattered light emitted from the top of the clouds. On the other hand, the ground-based European Cooperation for Lightning Detection (EUCLID) makes use of the low-frequency electromagnetic signals generated by
lightning discharges to locate those accordingly. The objective of this work is to quantify the similarities and contrasts between the latter two distinct lightning detection technologies by comparing the EUCLID cloud-to-ground strokes and intracloud pulses to the ISS-LIS groups, in addition to the correlation at the flash level. The analysis is based on the observations made during March 01, 2017 and March 31, 2019 within the EUCLID network and limited to 54° north. A Bayesian approach is adopted to determine the relative and absolute detection efficiencies (DE) of each system. It is found
that the EUCLID relative and absolute flash DE improves by approximately 10 % towards the center of the EUCLID network up to a value of 50.3 % and 69.4 %, respectively, compared to the averaged value over the full domain, inherent to the network geometry and sensor technology. On the other hand, the relative and absolute ISS-LIS flash DE over the full domain is 49 % and 68.9 %, respectively, and is somewhat higher than the values obtained in the centre of the EUCLID network. The behavior of the relative DE of each system in terms of the flash characteristics of the other reveals that the
greater the value the more likely the other system detects the flash. For instance, when the ISS-LIS flash duration is smaller or equal to 200 ms, the EUCLID relative flash DE drops below 50 %, whereas this increases up to 80 % for ISS-LIS flashes with a duration longer than 750 ms. Finally, the distribution of the diurnal DE indicates higher (lower) ISS-LIS (EUCLID) DE at night, related to an increased ISS-LIS:EUCLID flash ratio at night.

## 1 Introduction

Lightning processes in the cloud and from cloud-to-ground involve the formation of channels carrying tens of kiloamperes of electric current with temperatures as high as 30,000 K. Those processes are accompanied by intense radiation in the optical frequency range with the peak power typically being of the order of $10^9$ W (Christian et al., 1989). These optical emissions are a result of dissociation, excitation, and subsequent recombination of various atmospheric constituents as a result of the sudden intense heating, and primarily occur at discrete atomic lines. Satellite-based optical imagers operating in the visible
and near infrared frequency ranges record these optical emissions. The geolocation is carried out by using geometric projection of the images taken from space. In the seventies of last century different satellite programs started to use various





optical sensors to measure lightning, e.g., Vorpahl J.A. et al. (1970), Sparrow & Ney (1971) and Turman (1978). Due to the limited technology at this time, these satellite-based sensors had location accuracies of the order of hundreds of kilometers (due to the low spatial resolution of the optical imagers) and a detection efficiency of less than 2 %.

In 1995 the OV-1 (MicroLab 1) satellite carrying the optical transient detector (OTD) and in 1997 the TRMM satellite carrying the lightning imaging sensor (LIS) were launched. OV-1 orbited at an altitude of 750 km and TRMM at an altitude of 350 km and changed to 400 km after 2001 (Cecil et al., 2014). Therefore, the latter two satellites had a large field of view of 1300 x 1300 km and 600 x 600 km for OTD and LIS, respectively. Those optical imagers measure the signals emitted at 777.4 nm wavelength, associated with dissociation of molecular oxygen into atomic oxygen due to intense heating produced

by lightning processes. Data from such sensors typically consist of the time of occurrence of lightning event, latitude, and longitude. The radiance (brightness) for each pixel is also available, but interpretation of these measurements is complicated because the optical properties of the path between the emission and the measurement point vary. Since no relationship exists between the peak optical radiance measured by such sensors and peak currents of lightning events, estimated peak current and polarity of lightning events are not reported by satellite-based lightning sensors. OTD and LIS have a location accuracy

of about ten to a few tens of kilometers and a temporal resolution of a few milliseconds with better than 100 ms temporal accuracy, e.g., Boccippio et al. (2000). They detect emissions from both cloud and cloud-to-ground discharges but they cannot distinguish between them. Total flash detection efficiency for LIS during the day and night is estimated to be about 70 % and 88 % respectively and about 38 % and 52 % respectively for OTD, see Boccippio et al. (2002) and Cecil et al. (2014).

It is important to note that, similar to VHF lightning mapping systems, optical imagers are able to map the full spatial extent of flashes, although with poorer temporal and spatial resolution, and hence may be viewed as lightning mapping systems. Since these optical imagers on low earth orbiting satellites observe a given location on earth's surface for a limited time; typically around 90 seconds to a few minutes, they can only take snapshots of thunderstorms and cannot monitor them as they develop and evolve.

Generally, for all applications of lightning data it is important to know the performance of the employed lightning location system (LLS). The performance characteristics of lightning locating systems are determined by their ability to geolocate lightning events with high location accuracy (LA), high detection efficiency (DE), with low false detections and to report various other features of the lightning discharge correctly. Different methods or a combination of methods may be used to validate the performance characteristics of different types of lightning locating systems – see Nag et al. (2015). To get

information about performance variations over large spatial regions of ground based LLS, data of those systems were compared to data from TRMM-LIS. During the last years several papers provided additional insights in the performance of ground based networks with such an analysis, e.g., the WWLLN (Rudlosky & Shea, 2013), the NLDN (Zhang et al., 2016), the ENTLN (Rudlosky, 2015), the ATDnet (Enno et al., 2018) and the GLD360 (Rudlosky et al., 2017). One have to keep in mind that during the last years the performance of the ground based networks improved significantly and therefore the

analyses of data between 2008 and 2014 may not provide information about the current LLS performance.



In April 2013 it was decided that a LIS, built as the flight spare for the TRMM satellite, should be put to the International Space Station (ISS). The data of this sensor, called ISS-LIS, was analyzed in Erdmann et al. (2019) for the time period March 2017 to March 2018. They compared ISS-LIS data to the low-frequency LLS of Météorage and the lightning mapping array SAETTA (Coquillat et al., 2019) over Corsica.

In this paper the performance of EUCLID (EUropean Cooperation for LIghtning Detection), a ground based LLS, will be evaluated relative to the ISS-LIS data. This work is timely, given that the Meteosat Third Generation (MTG), which has a lightning imager (LI) on board, is going to be launched in 2 years.

## 2 Data

### 2.1 EUCLID

Starting in 2001 the European Cooperation for Lightning Detection (EUCLID) geolocates cloud-to-ground (CG) strokes and intracloud (IC) pulses through a combination of time-of-arrival (TOA) and direction finding (DF) techniques. The EUCLID cooperation is special in the sense that it combines raw sensor data in real-time of independent lightning location systems — either managed by National Meteorological Services (NMS) or by private companies — within a single processor. This is possible since all of the sensors operate in the same low-frequency (LF) range and are from the same manufacturer, i.e.,
Vaisala. The central processor of EUCLID adopts individually calibrated sensor gains and sensitivities to account for any local sensor site conditions. Those values can differ from the ones used by the local LLS provider due to the implicit higher redundancy in EUCLID as a result of the participation of additional sensors located outside the national borders in a neighboring country. Hence, it assures that the resulting lightning data are of high and nearly homogeneous quality throughout Europe. The performance of EUCLID has been frequently tested over the years in terms of its LA, DE and peak
current estimation. Those performances have been determined either from direct lightning measurements at the Gaisberg Tower (GBT) (Diendorfer et al., 2009), Peißenberg tower in Germany (Heidler & Schulz, 2016) and Säntis tower in Switzerland (Romero et al., 2011; Azadifar et al., 2016) as well as from video and E-field records collected in different regions within Europe (Poelman et al., 2013; Schulz et al., 2016). The current LA is in the order of 100 m based on the location error directly measured at the GBT and based on video and E-field recordings within the majority of the network.
The DE for negative CG strokes/flashes reaches 70/96 % based on GBT data and is determined to be 84/98 % using video and E-field records. On the other hand, the DE for positive events is greater than 84 % and 87 % for strokes and flashes, respectively (Schulz et al., 2016). Finally, IC DE has been validated during the HyMeX experiment (Ducrocy et al., 2013; Defer et al., 2015) in the south of France (Schulz et al., 2014; Pédeboy et al., 2014). For this purpose, EUCLID observations were matched to the observations made by the Lightning Mapping Array "HyLMA". It is found that the DE of isolated IC
flashes, i.e., pure IC flashes without any CG stroke in it, has a large variation ranging from 10 % up to 67 % from one thunderstorm to another. This variability is mainly attributed to differences in the vertical extent of the IC flash and to the flash rates during a storm. Regarding the peak current estimates, EUCLID tends to overestimate those slightly with respect to



the currents measured at the GBT with a median error of 4 %. More information regarding the performance and observations by the EUCLID network are found in Schulz et al. (2016) and Poelman et al. (2016).

## 2.2 ISS-LIS

The Lightning Imaging Sensor (LIS) aboard the International Space Station (ISS) is identical to the LIS used on the Tropical Rainfall Measuring Mission (TRMM) satellite which was operational from 1999 to 2015. LIS on ISS was installed in February 2017 with an intended mission lifespan of two years and collects lightning data from that point onward from a low earth orbit (LEO) at an altitude of about 408 km which is similar to the altitude of the TRMM satellite after 2001. The LIS sensor combines a wide field-of-view (fov) lens with a narrow-band interference filter of 1 nm centered on the strong oxygen triplet emission line at 777.4 nm. In addition, LIS employs an optical staring imager composed of a 128 x 128 charged coupled device (CCD) array with a sampling rate of approximately 500 frames per second. Although latitudinal coverage is expanded poleward to 54° due to a larger orbit inclination (55° instead of 35° for the TRMM satellite) the performance characteristics of ISS-LIS are similar to that of TRMM-LIS. This means that the electrical activity within thunderstorms is detected with a resolution of 4 km at nadir and increases somewhat towards the edge of the measurement region, with a swath width of about 650 km of the Earth's surface. Hence, due to the continuous movement of the ISS with an orbital speed of approximately 7 km/s, lightning observations over a specific region lasts no longer than 90 seconds per overpass. When a lightning discharge occurs, the optical signal scatters throughout the cloud. In almost all of the cases, except in the unlikely case the cloud is extremely optically thick, this results in an extended area being light up on the top of the cloud when viewed from space. At the moment a pixel on the CCD array receives this optical pulse, the signal is compared to the dynamically changing detection background threshold. Once this threshold is exceeded, the processor identifies this illuminated pixel as a LIS event. It is important to note that a LIS event has no counterpart when compared to the observations made by a ground-based LLS such as EUCLID. However, the collection of LIS events from adjacent pixels during the same 2 ms frame integration time, defined as a LIS group, is comparable with either a CG stroke or cloud pulse. Note that the ISS-LIS group location is the radiance-weighted centroid of all the events within the respective group (Mach et al., 2007). In its turn, groups are clustered within a flash when the spatial and time criteria of 5.5 km and 330 ms, respectively, are met. In contrast to EUCLID, LIS is not able to distinguish between CG and IC lightning. Nevertheless, Boccippio et al. (2002) estimated an upper bound for the TRMM-LIS total flash DE of 88 ± 9 %.

In this work, we make use of the non-quality controlled ISS-LIS dataset (Publication date: 2019-08-19, Version 1, Processing level 2) made available by the NASA Global Hydrology Resource Center DAAC. This includes information on geolocated and time-tagged lightning events, orbit statistics and metadata. For more in depth information on the LIS instrument, the interested reader is referred to Christian et al. (1989), Blakeslee et al. (2014) and Blakeslee & Koshak (2016).





## 3 Methodology

In this paper EUCLID and ISS-LIS lightning observations are correlated using data in between March 01, 2017 until March
130    31, 2019, as observed within the EUCLID domain and limited to 54° north. ISS-LIS detects optically bright discharges, such
as return strokes and in-cloud discharges inducing a rapid change in the electric field (Goodman et al., 1988). Those rapid
changes in the electric field are exactly the features detected by EUCLID. Hence, the fundamental unit of ISS-LIS, i.e.,
groups, and EUCLID, i.e., CG strokes and IC pulses, largely corresponds to the same physical process and is therefore
directly comparable (Bitzer et al., 2016). Additionally, the comparison will be performed as well on the artificial derived
135    flash level.

The approach taken in this work has been applied and described in detail in Rubinstein (1994) and Bitzer et al. (2016), in
which a probabilistic method is used to estimate the relative and absolute detection efficiencies of both systems under
investigation. The concepts are briefly defined below. Neither EUCLID nor ISS-LIS observe all of the lightning activity that
actually occurred at a given moment in time. Hence, let S be the set of all occurred lightning discharges and A and B the set
140    of discharges detected by ISS-LIS and EUCLID, respectively, as illustrated in Figure 1. Note that it is possible that both of
the systems contain some false alarm detections and therefore fall outside S. However, those false alarms constitute roughly
1 % of the total amount of events detected by EUCLID (Poelman et al., 2017), whereas the false event rate requirement for
LIS is set to be less than 5 %. Hence, the latter has a minor influence on the final outcome. The system dependent relative
detection efficiencies can be expressed as:


$$P(A \mid B) = \frac{n_A \bigcap n_B}{n_B} \tag{1}$$

$$P(B \mid A) = \frac{n_A \bigcap n_B}{n_A} \tag{2}$$

with $n_A$ and $n_B$ the amount of discharges detected by system A and B, respectively, and $n_A \bigcap n_B$ the intersection, containing
discharges detected by both systems. Thus P(A|B) represents the conditional probability that LLS A detects a discharge
relative to LLS B, and vice versa in case of P(B|A). In addition, the true detection efficiency, for example of system A, reads
as:

$$P(A) = \frac{n_A}{n_S} \tag{3}$$

However, the actual amount of occurred discharges $n_S$ is not known a priori. Therefore, the estimated absolute detection
efficiencies (ignoring false detections) can be calculated in the following way:



$$P(A) = \frac{n_A}{n_S} \leq \frac{n_A}{n_A + n_B - n_A \bigcap n_B} \tag{4}$$

$$P(B) = \frac{n_B}{n_S} \leq \frac{n_B}{n_A + n_B - n_A \bigcap n_B} \tag{5}$$

Since the number of detections in S, $n_S$, is larger than the unique set of combined discharges in A and B, the estimated absolute DE is an upper limit for the true detection efficiency. However, to precisely calculate the above detection efficiencies only those EUCLID discharges that occurred within the ISS-LIS fov, $n_B$, need to be taken into account. To this end, the corner points of two consecutive ISS-LIS fovs, separated by roughly 35 seconds, are extrapolated to every second to increase accuracy. Then, for each second the EUCLID detections are extracted within the respective fov. As an example, the

ISS-LIS groups and EUCLID CG strokes and IC pulses are plotted on top of the ISS-LIS fov in Fig. 2. This is the biggest difference compared to the future MTG-LI observations from a geostationary orbit. Next, the individual EUCLID CG strokes and IC pulses are correlated in time and space with the ISS-LIS groups in order to retrieve the amount of overlapping detections. A match is found when the temporal and spatial criteria of 10 ms and 20 km, respectively, are fulfilled. Those particular criteria have been used in similar inter-comparison studies such as Franklin (2013), Bitzer et al. (2016) and Zhang

et al. (2016; 2019). Note that only one LIS group can be matched to a single EUCLID discharge and vice versa. At the flash level, matching is somewhat more complicated due to the fact that EUCLID and ISS-LIS have their own specific flash clustering algorithms. For the flash analysis EUCLID strokes/pulses are matched to ISS-LIS groups using a larger temporal (100 ms) and spatial (30 km) criterion to account for the fact a flash can consist out of different discharges spread over some time interval. Subsequently, the strokes/pulses and groups are traced back to the respective flash it belongs to. Thus a

matched flash can have one or multiple matched discharges or groups. Note that since the flash grouping algorithms between EUCLID and ISS-LIS are different, the matched flash count $n_A \bigcap n_B$, used in Eq. 4 and 5, is slightly different depending on the network, even though the matched count of EUCLID discharges and ISS-LIS is the same.

## 4 Results

### 4.1 EUCLID stroke/pulse and ISS-LIS group level

In Fig. 3 the spatial distribution of the IC:CG ratio, observed by EUCLID, is plotted at the level of the IC pulses and CG strokes, as well as at the flash level. Only data within the EUCLID domain as indicated by the dashed polygon and cut off at 54° north to account for the ISS-LIS latitudinal coverage are used for quantitative analysis in this work. The geographic spread does not reflect the actual IC:CG occurrence within Europe, but mainly highlights areas where EUCLID is more capable detecting IC activity over others due to sensor technology. Not surprisingly, the highest IC:CG ratios are found in

regions where the baseline between LS700x sensors is small and drops off towards the south and east of the domain where




mainly IMPACT sensors were installed during the period of investigation. Additionally, during this time period, in the south of Italy and in Spain significant communication problems deteriorated the results. The mean IC:CG stroke (flash) ratio over the entire region is 2.6 (1.9) and increases to 4.1 (2.8) within the rectangle highlighted in white. The rectangle highlighted in white in Fig. 3 will be referred to as the centre of the EUCLID network throughout the paper. The mean IC:CG flash ratio in

the centre of the network is comparable to the values observed by the U.S. National Lightning Detection Network (NLDN) in various parts throughout the contiguous United States as presented by Medici et al. (2017). Since EUCLID observes most of the IC pulses in the centre of the network, in the remainder of the paper results will be presented for the full domain, as well as for the centre where indicated.

The distance offset $\Delta d$ in one kilometre intervals between matched EUCLID detections and ISS-LIS groups is indicated in

Figure 4 and expressed in percentage of occurrence. A steep rise is observed up to 2-3 km, followed by a steady decrease for the larger distance offsets. The mean (median) location difference is 5.7 km (4.8 km) corresponding to approximately two pixels in the ISS-LIS CCD imager. This result is in line with previous findings as presented in Bitzer et al. (2016) and Zhang et al. (2019) who compared LIS group locations with comparable ground-based LLS. Towards the centre of the EUCLID network, the mean and median distance offset drops by 180 m and 200 m, respectively. This is not surprising since shorter

baselines amongst the sensors in this region and the use of sensor based onset time calculation lead to a better location accuracy and hence better correspondence with the LIS group positions.

Similar as with the distance offset, the timing differences $\Delta t$, calculated here as $t_{\text{ISS-LIS}} - t_{\text{EUCLID}}$, can be calculated between matched discharges detected by both systems. The distribution of the time offset between matched ISS-LIS groups and EUCLID pulses is indicated in Figure 5. A negative (positive) value indicates that the ISS-LIS group occurred earlier (later)

than the EUCLID match. It is found that the distribution peaks sharply around ±1 ms, with a mean (median) time offset of -1.3 ms (-0.6 ms). Thus, on average an ISS-LIS group occurs first. Nonetheless, the majority of the timing differences fall inside the ISS-LIS timing accuracy set by the frame integration time of 2 ms. Unlike for the distance difference, the time offset does not differ within the EUCLID domain.

The estimated peak current of matched EUCLID CG strokes (solid line) and IC pulses (dashed line) are correlated with the

ISS-LIS group radiance in Figure 6. Note that positive discharges with peak currents smaller than 10 kA are likely to be misclassified as CG strokes because those are more likely to be of intracloud nature (Cummins et al., 1998; Wacker & Orville, 1999a;b; Jerauld et al., 2005; Orville et al., 2002; Biagi et al., 2007). Hence, positive CG strokes below 10 kA are all categorized as IC pulses and therefore no data for positive CG below 10 kA exist. Additionally, the largest positive IC pulse in the EUCLID dataset has an estimated peak current of 28.8 kA, limiting the positive IC pulse curve in the plot. In general,

higher peak current signals observed by EUCLID correspond with higher ISS-LIS group radiances. At larger absolute peak current values, i.e., $|I_p| \geq 20$ kA, the correlation becomes more variable. However, the latter is an artefact of the sample size, as indicated by the grey curve in the plot.

Finally, the relative and absolute detection efficiencies can be calculated using the formulas described in Sect. III. It follows that the ISS-LIS group relative DE, i.e., P(ISS-LIS|EUCLID), is 34.3 % , while it is 8.1 % in case of P(EUCLID|ISS-LIS)


over the full domain. In the centre of the EUCLID network P(ISS-LIS|EUCLID) drops slightly to 33.9 %, while P(EUCLID|ISS-LIS) increases to 11.3 %. Those values are comparable with the relative detection efficiency values presented in Zhang et al. (2016), correlating NLDN detections with TRMM-LIS observations in 2013 over the CONUS, although P(TRMM-LIS|NLDN) was somewhat higher at 52.9 %. A possible explanation of the lower P(ISS-LIS|EUCLID) found in this study compared to P(TRMM-LIS|NLDN) in Zhang et al. (2016) is the advancement of the LS700x technology

in detecting IC pulses in between 2013 and the period 2017-2019 used in this study. Thus, in this work the EUCLID dataset contains more IC pulses, which have no counterpart in the ISS-LIS observations. Upper bound for the ISS-LIS absolute detection efficiency, P(ISS-LIS), is 86.5% and 82.0% in the full domain and the centre of the network, respectively, while P(EUCLID) is 20.5 % and 12.9 %, respectively.

**4.2 Flash level**

The spatial distribution of the EUCLID and ISS-LIS flash counts is indicated in the upper plots of Figure 7, while the lower plots show the geographic spread of the absolute flash detection efficiencies P(EUCLID) and P(ISS-LIS). The overall spatial behavior of the flash counts is similar between the two detection systems. However, the biggest difference in flash counts is found to be outside the center of the EUCLID network, especially over Spain, Italy and the Mediterranean Sea, where ISS-

LIS outperforms EUCLID in terms of the amount of detections. For those regions the same comment applies as for the IC:CG ratio above, namely the communication problems in those regions. P(ISS-LIS|EUCLID) is 49.0 % overall and 48.5 % in the center of the network, while Zhang et al. (2016) found that P(TRMM-LIS|NLDN) to be 68.3 %. Similar as in Section A, assuming that TRMM-LIS and ISS-LIS have the same performance, the significant smaller average sensor baseline of the EUCLID network compared to the NLDN leads to an increased IC flash component in the EUCLID observations without

any counterpart in the ISS-LIS flash observations, hence leading to a lower relative ISS-LIS DE in this work. On the other hand, P(EUCLID|ISS-LIS) increases from 39.8 % overall to 50.3 % in the center of the network. The latter result is in line with the 48.7 % as found in Zhang et al. (2016). An extra region is outlined in Figure 7 covering Corsica, corresponding with the region investigated by Erdmann et al. (2019) in which ISS-LIS flashes are matched against those observed by the ground-based Météorage network during a 1-year period from March 1, 2017 until March 20, 2018. Applying analogue flash

matching criteria as in Erdmann et al. (2019) over the Corsica area, P(ISS-LIS|EUCLID) becomes 61.9 % in this study, being identical to the 62.4 % found in Erdmann et al. (2019). Erdmann et al. (2019) found P(Météorage|ISS-LIS) to be 83.3 % over Corsica while in this work P(EUCLID|ISS-LIS) is 52.5 %. This difference however is attributed to the fact that Météorage benefits from additional LS700x sensors around Corsica in addition to the sensors in EUCLID, hence improving its IC detection efficiency. The absolute DE of ISS-LIS, P(ISS-LIS), is ≤68.9 % overall and drops to 63.4 % in the center of

the EUCLID network, being somewhat smaller than the 81.5 % in Zhang et al. (2016). P(EUCLID) is 59.4 % overall and increases to 69.4 % in the center of the network, while it is 58.2 % in Zhang et al. (2016). Smaller baselines in the center of the EUCLID network lead to a better P(EUCLID), whereas the full domain includes oceans and regions with larger



baselines.. Additionally, note that in Zhang et al. (2016), the NLDN observations used were restricted to the areas where the NLDN detection efficiency is highest. Hence, P(EUCLID) of 69.4 % in this work should be compared to the 58.2 % in Zhang et al. (2016). From Figure 7 it is found that P(ISS-LIS) is highest outside the center of the EUCLID network, while it is the opposite in case of P(EUCLID). Note that, contrary to what is found in this study, the absolute DE for ISS-LIS should be uniform over the entire region, since it is highly unusual to expect a geographic dependence. It is believed that the spatial dependence is related to the limit of the Bayesian algorithm using only two networks. Probably making use of additional networks would make it disappear.

Average characteristics of all ISS-LIS flashes, those observed (matched) and not observed (unmatched) by EUCLID are listed in Table 1, while Figure 8 provides relative detection efficiency as a function of those characteristics. Note that the ISS-LIS flash characteristics as in Table 1 do not differ much throughout the EUCLID domain. Therefore those latter values are averages over the full domain. From Table 1, it is found that the duration of ISS-LIS flashes is longer by a factor of 1.5 for those that have a match with a EUCLID flash compared to those not observed by EUCLID. Additionally, Fig. 8a demonstrates an increasing trend in the EUCLID flash DE with increasing flash duration. For flashes lasting longer than about 750 ms, the flash DE is stable around 60 % averaged over the full domain and increases further by about 20 % in the center of the EUCLID network. On average, ISS-LIS flashes consist of 8.6 groups, while matched flashes contain 10.7 groups and drops to 7.2 groups per flash for the unmatched flashes. From Fig. 8b a sharp increase is observed in the EUCLID flash DE for ISS-LIS flashes with up to about 10 groups, followed by a moderate rise in flash DE for flashes up to 30 groups. ISS-LIS flashes containing more than 30 groups have the highest probability to match a EUCLID flash, resulting in a EUCLID flash DE between 60 %-80 %. A similar behavior is found when looking at the maximum number of events per group (MNEG), which has been plotted as well in Fig. 8b. Examining the behavior as a function of amount of events per flash, an analogue tendency is observed compared to the previously examined behavior of groups per flash. Flashes containing 75 events or more boost the EUCLID flash DE up to 80 % in the center of the EUCLID network (see Figure 8c).

Fig. 8d demonstrates an identical behavior of the EUCLID flash DE as a function of ISS-LIS flash area and maximum group area (MGA) with larger flash DE for those flashes that have a larger optical footprint. Not surprisingly, the flash DE is proportional to the ISS-LIS flash radiance and increases sharply up to an ISS-LIS flash radiance of about 1000 µJ/sr/m$^2$/µm, as demonstrated in Figure 8e. The results presented above are similar to the ones found by Rudlosky et al. (2017) evaluating the GLD360 DE with respect to TRMM-LIS flash characteristics.

Conversely, the ISS-LIS flash DE can be linked to EUCLID flash characteristics, such as its flash duration, multiplicity and area. Note that to calculate the area of a EUCLID flash the minimum and maximum latitudinal and longitudinal coordinates within the flash are used. Results hereof are presented in Table 2 and distributions drawn in Figure 9. From Fig. 9a it follows that EUCLID flashes with a duration lower than 200 ms have the least chance to match an ISS-LIS flash, resulting in a ISS-LIS flash DE below 60 %. EUCLID flashes lasting longer than 800 ms have an 80 % probability to be matched to an ISS-LIS flash. On average matched EUCLID flashes last twice as long compared to the unmatched flashes. Furthermore, EUCLID flashes have an average multiplicity of 2.4, while this is 2.9 (2.1) for the flashes that match (unmatch) an ISS-LIS


flash. For EUCLID flashes with a multiplicity lower than 3, the ISS-LIS flash DE lies between 40-60 %, and increases up to about 70 % for EUCLID flash multiplicities greater than 4, as evidenced in Fig. 9b. Although matched EUCLID flashes have an average area of 25.4 km$^2$, twice the size of the unmatched EUCLID flashes, the behavior of the ISS-LIS flash DE as a

function of EUCLID flash area is not as pronounced, compared to the distributions as a function of EUCLID flash duration and multiplicity.

The diurnal behaviour of the absolute detection efficiency for ISS-LIS as well as for EUCLID at the level of groups/strokes and flashes is plotted in Figure 10. In addition, the ISS-LIS:EUCLID flash ratio is plotted as well. The absolute ISS-LIS group and flash DE clearly shows an increase at night, compared to daytime, whereas the opposite is noticeable in case of

the absolute EUCLID DE. This effect is more pronounced for the flashes. At the level of the flashes, the EUCLID absolute DE drops from about 60 % during [6, 18] UTC to about 50 % within [18, 6] UTC, whereas the ISS-LIS absolute DE increases from about 70 % in between [6, 18] UTC to 75 % within [18, 6] UTC. This behavior is clearly linked to the increased capacity of ISS-LIS to detect more flashes during night time, whilst during the day the flash ratio of ISS-LIS:EUCLID is much closer to one.

## 5 Conclusions

There exists a multitude of different technologies to detect and locate the electrical activity in thunderstorms, whether on a local, continental or global scale. This leads to various sets of observations of the same phenomenon. Hence, depending on the requirements, e.g., spatial accuracy and/or extent, the user can favor one system over the other. Nevertheless, it is important to investigate the similarities and differences among different systems. Incited by the forthcoming launch of the

Meteosat Third Generation geostationary satellites, with onboard the Lightning Imager, this study aims at comparing for the first time over a large area in Europe the lightning observations from the ground-based EUCLID network to the optical signals detected by the space-based ISS-LIS. The analysis is based on the lightning activity recorded during March 01, 2017 and March 31, 2019 within the EUCLID network and limited to 54° north. In this study the EUCLID cloud-to-ground strokes and intracloud pulses are compared to the ISS-LIS groups, in addition to the correlation at the level of the flashes of

both systems. Besides measuring the temporal and spatial differences between matched observations a Bayesian approach is adopted to determine the relative and absolute detection efficiencies (DE) of each system.

It is found that the matched EUCLID strokes/pulses and ISS-LIS groups are separated by a median distance of 4.8 km, corresponding to approximately two ISS-LIS pixels in the CCD imager. A negative median time difference, $t_{ISS-LIS} - t_{EUCLID}$, between matched discharges of 0.6 ms is well within the time accuracy of ISS-LIS and indicates that on average the ISS-LIS

group occurs first. On the other hand, higher peak current signals observed by EUCLID correspond with higher ISS-LIS group radiances. The ISS-LIS group relative DE is 34.3 % and drops slightly to 33.9 % in the centre of the EUCLID network. The latter values are in contrast to the much lower EUCLID stroke/pulse relative DE values of 8.1 % and 11.3 % over the full domain and in the centre of the network, respectively. This is related to the higher amount of ISS-LIS groups

detected compared to total amount of EUCLID strokes and pulses. A similar behaviour is observed for the upper bound of the absolute DE with values of 86.5% and 20.5 % for ISS-LIS and EUCLID, respectively.

On the level of the flashes, the relative ISS-LIS DE is rather homogeneous over the entire EUCLID network with a value of 49.0 % with the highest values observed at the edge of the EUCLID network, whereas the EUCLID relative DE increases from 39.8 % overall to 50.3 % in the center of the network, and is related to the increased EUCLID IC:CG flash ratio towards the center of the network. The upper bound of the absolute DE for ISS-LIS drops from 68.9 % overall to 63.4 % in 325 the center of the EUCLID network, whereas in case for EUCLID this value increases from 59.4 % overall to 69.4 % in the center of the network.

The behavior of the relative DE of one system in terms of the flash characteristics of the other reveals that the greater the value, the more likely the other system detects the flash. For instance, it is found that the duration of ISS-LIS flashes is longer by a factor of 1.5 for those that have a match with a EUCLID flash compared to those not observed by EUCLID, 330 while the other way around matched EUCLID flashes last twice as long compared to the unmatched flashes.

Finally, the absolute ISS-LIS group and flash DE clearly shows an increase at night, compared to daytime, whereas the opposite is noticeable in case of the absolute EUCLID DE. This behavior is related to the increased capacity of ISS-LIS to detect more flashes during night time, whilst during the day the flash ratio of ISS-LIS:EUCLID is much closer to one.

**Author contribution**

DRP processed and analyzed the different datasets used in this paper, whereas WS provided feedback along every step in the process. Writing the manuscript was a joint effort of DRP and WS.

**Competing interests**

The authors declare that they have no conflict of interest.

**Acknowledgement**

Special thanks go to Zhang D. and Erdmann F. for benchmarking the scripts used in this work.

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

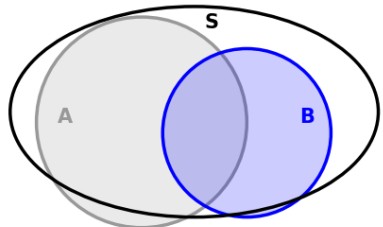

**Figure 1. A Venn diagram illustrating set S as total lightning discharges, whereas A and B are a set of discharges observed by independent systems. The intersection of A and B is composed of discharges detected by both A and B. Note that there can be a small portion of false alarm discharges detected by either system, that occur outside set S.**



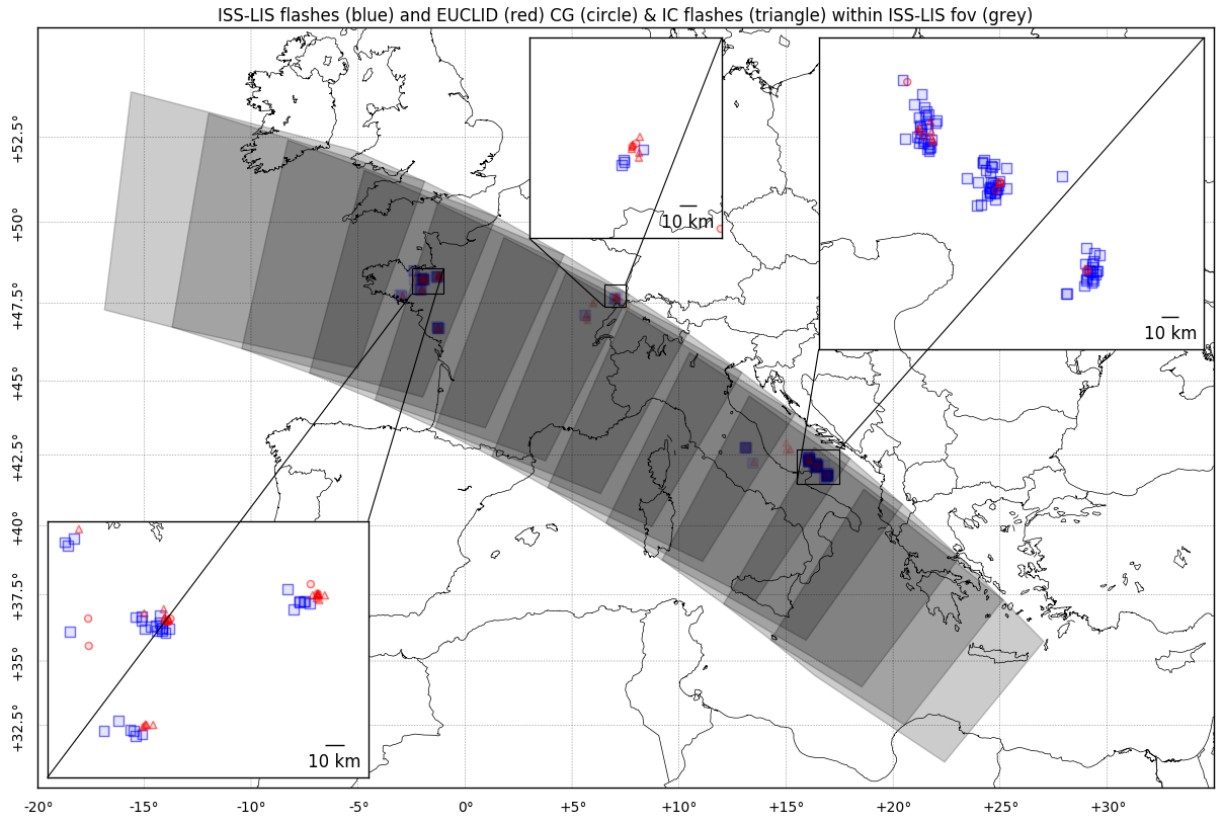

**Figure 2.** Example of a ISS-LIS track over Europe. The different field of views (fov) are in gray. Zoom-in show the ISS-LIS flashes (blue) together with the CG and IS flashes observed by EUCLID.








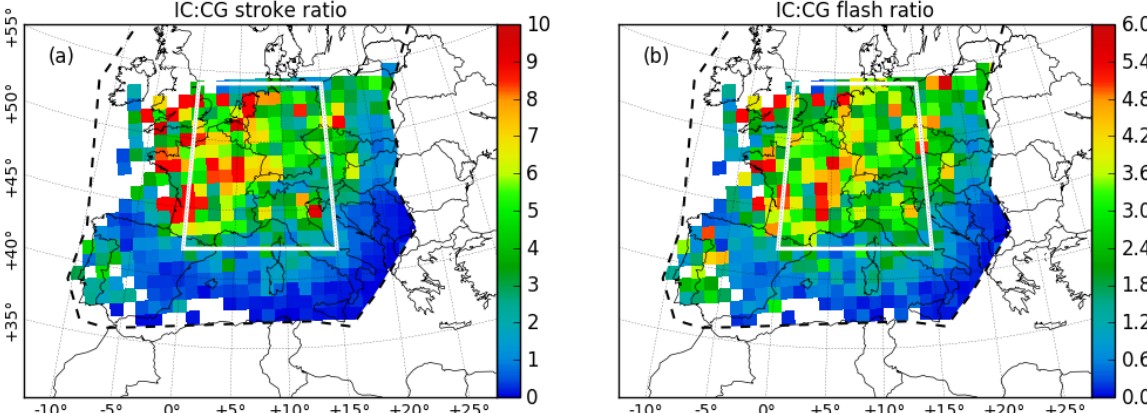

**Figure 3. IC:CG (a) stroke and (b) flash ratio within the EUCLID domain and cut-off at 54° according to the ISS-LIS maximum latitudonal observations. The rectangle highlighted in white is the area with the highest IC:CG ratio and is referred to as the centre of the EUCLID network throughout the paper.**






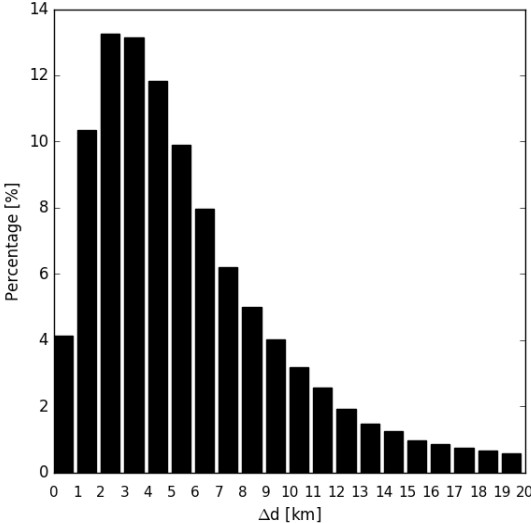

Figure 4. Distance offset between matched ISS-LIS groups and EUCLID strokes.


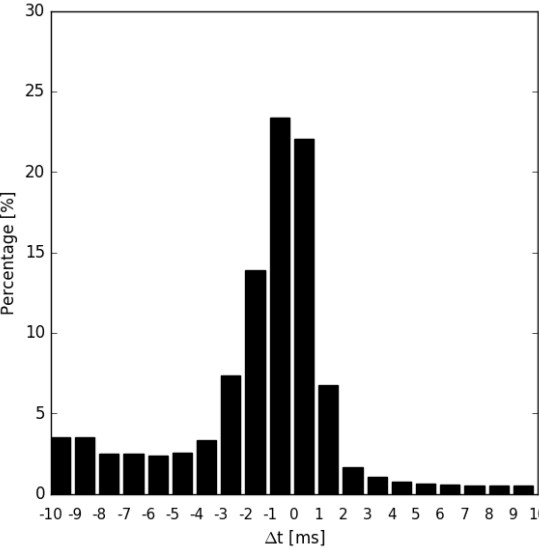

Figure 5. Time offset between matched ISS-LIS groups and EUCLID strokes. A negative/positive value indicates that the ISS-LIS group occurred earlier/later than the EUCLID stroke match.





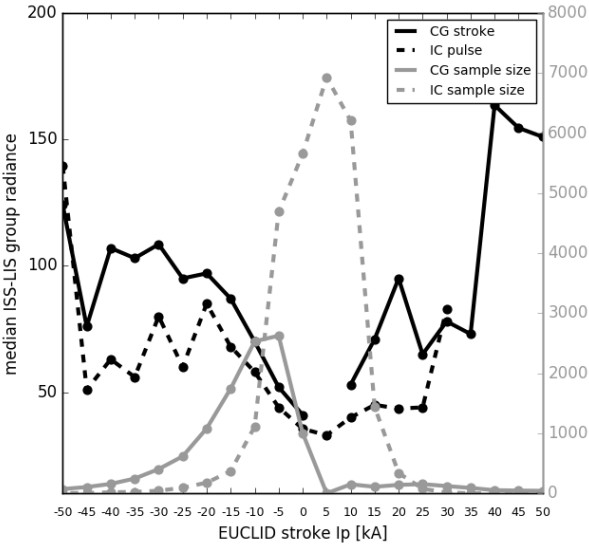

**Figure 6. The estimated peak current of matched EUCLID CG strokes (black) and IC pulses (grey) are correlated with ISS-LIS group radiance. In general, higher peak current signals are correlated with higher ISS-LIS radiances.**




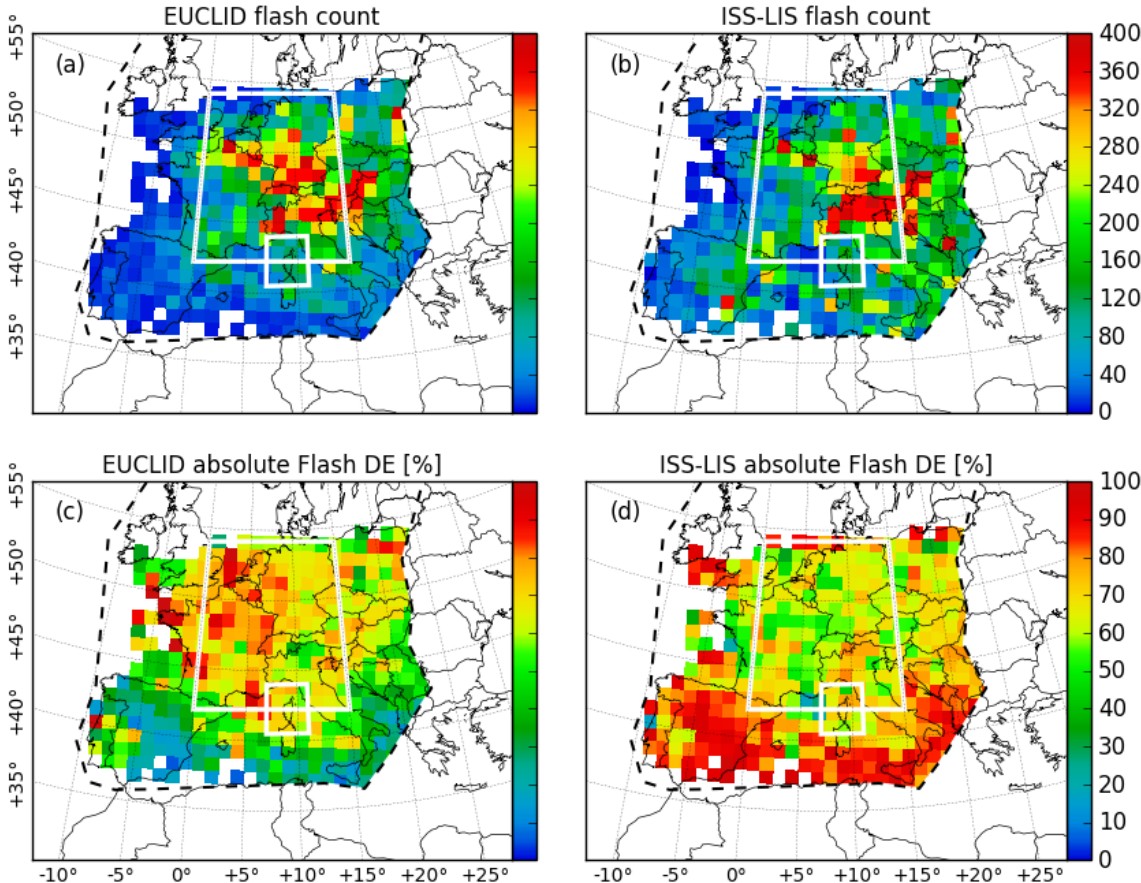

**Figure 7.** EUCLID and ISS-LIS flash densities are depicted in a) and b), respectively. Plots c) and d) show the spatial distribution of the absolute detection efficiencies. The large rectangle highlighted in white is the self-defined centre of the EUCLID network, whereas the smaller white rectangle highlights the area used in Erdmann et al. (2019) against which the results in this paper are compared to.







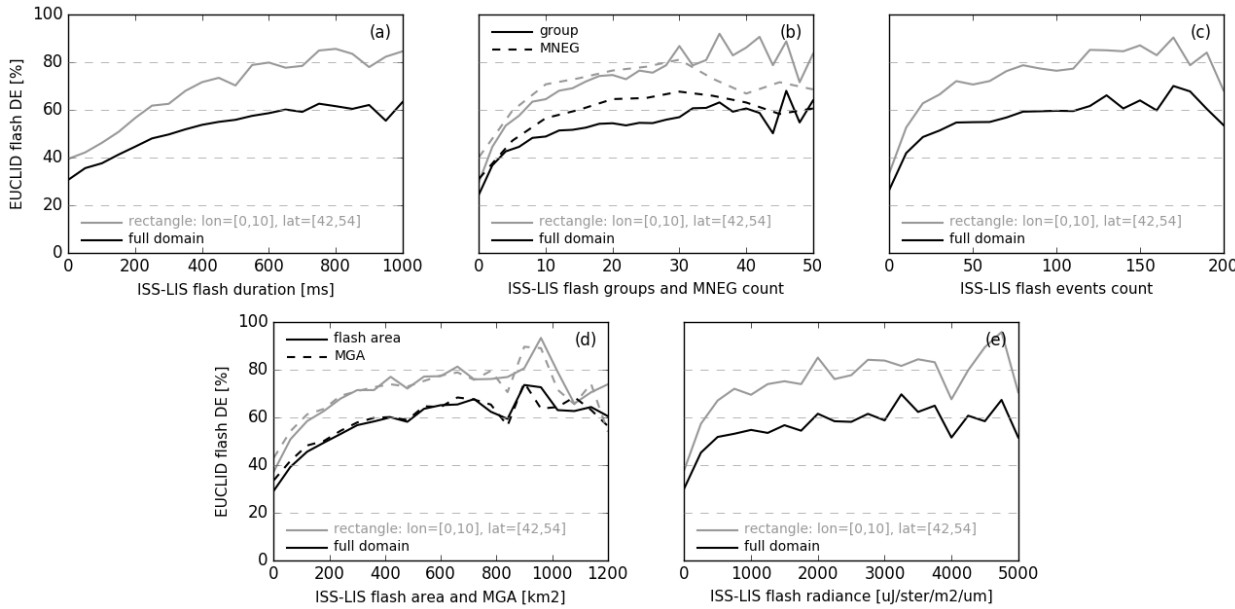

**Figure 8. EUCLID flash DE as a function of ISS-LIS flash characteristics such as a) flash duration, b) amount of groups in flash and maximum number of events in a group, c) amount of events, d) flash area and maximum group area and e) flash radiance. In black are the results for the full domain, whereas in gray the results are plotted within the rectangle: lon[0,15], lat[42,54]**


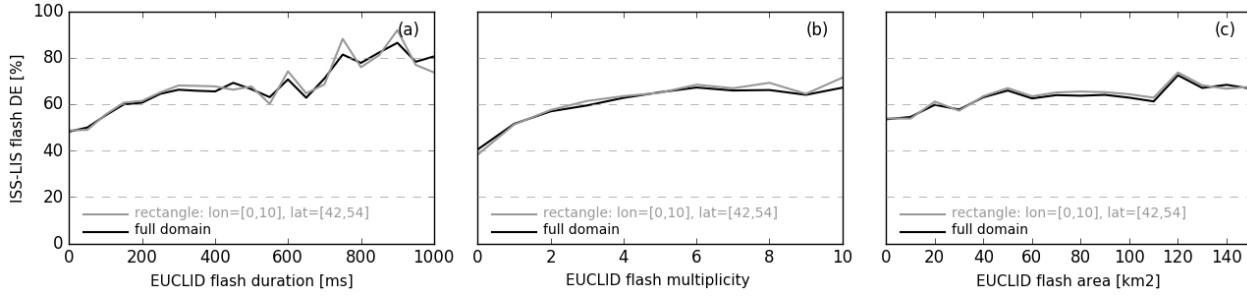

**Figure 9. ISS-LIS flash DE as a function of EUCLID flash characteristics such as flash a) flash duration, b) multiplicity and c) area. In black are the results for the full domain, whereas in gray the results are plotted within the rectangle: lon[0,15], lat[42,54]**



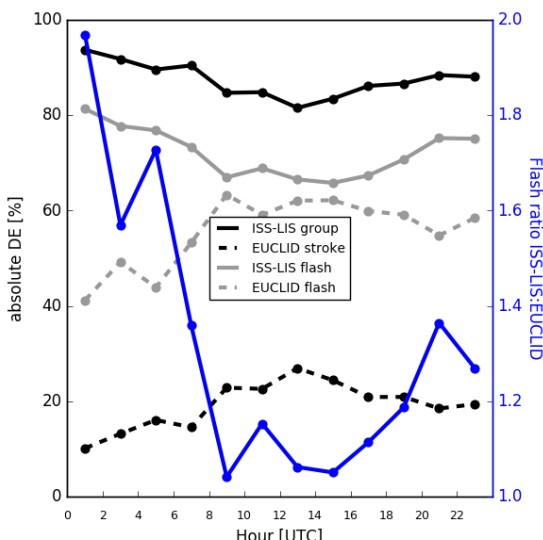

**Figure 10. Diurnal distribution of the absolute detection efficiencies.**


TABLE 1: Average characteristics of all ISS-LIS flashes, those observed by EUCLID (matched), and those not observed by EUCLID (unmatched).

| | ISS-LIS | Matched | Unmatched |
|---|---|---|---|
| Events [count] | 28.5 | 38.2 | 22.1 |
| Groups [count] | 8.6 | 10.7 | 7.2 |
| Duration [ms] | 210 | 267.4 | 172.2 |
| Area [km$^2$] | 172.4 | 219.8 | 141.0 |
| MNEG [count] | 6.8 | 8.3 | 5.8 |
| MGA [km$^2$] | 146.4 | 186.9 | 119.6 |
| Radiance [μJ/sr/m$^2$/μm] | 579 | 784 | 443 |

MNEG = maximum number of events per group
MGA = maximum group area





TABLE 2: Average characteristics of all EUCLID flashes, those observed by ISS-LIS (matched), and those not observed by ISS-LIS (unmatched).

|  | EUCLID | Matched | Unmatched |
|---|---|---|---|
| Duration [ms] | 102.9 | 136.9 | 70.5 |
| Multiplicity* | 2.4 | 2.9 | 2.1 |
| Area [km$^2$] | 18.6 | 25.4 | 12.1 |

*Multiplicity here means the number of strokes, pulses or the sum of both in a pure CG, IC, or hybrid flash, respectively.




