# Peer review of "Comparing lightning observations of the ground-based EUCLID network and the space-based ISS-LIS"

_Atmospheric Measurement Techniques, 2019_

## Referee Comment (RC1)

Review: Comparing lightning observations of the ground-based EUCLID network and the space-based ISS-LIS (https://doi.org/10.5194/amt-2019-435)

This manuscript compares the satellite-based optical ISS-LIS and ground-based radio EUCLID observations of lightning. It is an important study in that it helps evaluate both ISS-LIS and EUCLID networks and could potentially contribute to future LMI studies on the Meteosat. This study was very thorough and the manuscript is well written. Below are some of my comments.

Major comments:

1. I know that the terminology used in our community is a little confusing. A LIS "event" is not equivalent to a EUCLID "event. "  I would suggest adding a sentence somewhere to emphasize that, so the readers wouldn't get confused. Something like use the capital Event as a LIS "event."

2. The ISS-LIS data used in this study are non-quality controlled. That could, to some degree, affect the results. I would suggest adding the reference of Blakeslee's talk at the 2019 GLM meeting (https://goes-r.nsstc.nasa.gov/home/meeting-agenda-2019), which showed that ISS-LIS is slightly less sensitive than TRMM, and have a few sentences discussing the possible impact.

3. In Line 206, it says that "on average an ISS-LIS group occurs first." This looks a little odd to me. LIS group is an accumulation of a 2 ms period and the LIS timestamp is adjusted to the middle of that frame, which is 1 ms earlier than the end of the frame (Bitzer and Christian, 2015). So my impression is that the radio signals should be detected earlier. Zhang et al. (2016) also shows similar results. Do you have an explanation why ISS-LIS group occurs first in here? Possible scenarios could be that the EUCLID didn't report the initial breakdown processes in the flashes, while LIS reported them. If that's the case, then what is the detection efficiency of EUCLID reporting initial breakdown processes? Is it known that EUCLID has a poorer DE for initial breakdown than NLDN? I would suggest the authors double-check the results. If the results are correct, then this could lead to some further discussions.

Minor comments:

1. Line 41: Brightness is fine, but the LIS radiance is actually energy density (Koshak, 2010).
2. Line 216: artifact
3. Line 228: Should it be 12.9% and 20.5%, respectively? The P(EUCLID) in the full domain should be less than in the centre of the network, correct?
4. The very first paragraph on Page 9 (line number around 255) that discussed the absolute DE of ISS-LIS. The fact that ISS-LIS is less sensitive than TRMM-LIS might also contribute to the findings here. I would suggest adding that in the paragraph.

Reference:

Bitzer, P. M. and Christian, H. J., 2015: Timing uncertainty of the lightning imaging sensor. J. Atmos. Oceanic Technol., 32(3), 453-460, https://doi.org/10.1175/JTECH-D-13-00177.1.

Koshak, W. J., 2010: Optical characteristics of OTD flashes and the implications for flash-type discrimination. J. Atmos. Oceanic Technol., 27, 1822–1838, https://doi.org/10.1175/2010JTECHA1405.1.

---

## Referee Comment (RC2) · Anonymous Referee #2 · 19 Feb 2020

See attached PDF file.

**Review of the manuscript**

**"Comparing lightning observations of the ground-based EUCLID network and the space-based ISS-LIS"**

This work presents a comparison between two lightning detection methodologies, a traditional ground-based system (EUCLID) and the innovative space-based ISS-LIS. A detailed comparison between the two different systems has been carried out using data collected from March 01, 2017 to March 31, 2019 within the area covered by EUCLID network.

The authors deal with a topic of relevant interest that full satisfy the scope of AMT journal. The paper is well written and has a linear and clear structure. The results have been discussed after a deep analysis performed through a Bayesian method and more classical approaches. I think that the paper may be published in AMT after the authors have addressed the following questions.

- **Introduction (lines 67-69):** in my opinion, the authors should provide more details about Erdmann et al. (2019) work, in which data from ISS-LIS where compared against observations from on-ground lightning detection networks. The authors should better emphasize the "added-value" of their work compared to the previous study just mentioned.
- **Data (2.1 Euclid).** Why did the authors choose the data from EUCLID network for their analysis? I suggest to provide clear and strong motivations about this choice. It is well known that in European area other lightning detection network are available, providing data about cloud-to-ground (CG) and intra-cloud (IC) flashes with high detection efficiency.
- **Data (2.2 LIS, Lines 124):** the authors stated that they used a non-quality controlled ISS-LIS dataset. I think that some clarifications are needed. What does mean "non-quality controlled"? How did the authors overcome this problem?
- **Methodology (Line 168):** I suggest to produce a Figure or a Table to support and justify these choices about spatial and temporal criteria.
- **Results:** to improve the quality of the presentation of the findings of this study, I propose to produce one or two additional tables.
- **Conclusions:** please add a brief discussion about the future implications of this work. I think it may have a good impact from different perspectives. Therefore, the conclusions section should not be limited to a summary of the main results.

Finally, I suggest to carefully check the paper to address some minor typos.

**Fig. 1.**

---

## Author Comment (AC1) · 30 Mar 2020

**REVIEWER 1**

**Major comments:**

1. I know that the terminology used in our community is a little confusing. A LIS "event" is not equivalent to a EUCLID "event". I would suggest adding a sentence somewhere to emphasize that, so the readers wouldn't get confused. Something like use the capital Event as a LIS "event."

=> Correct. Care must be taken to avoid confusion. Hence, at various places in the text "EUCLID events" is changed into "EUCLID discharges" and ISS-LIS "events" remain "events" where appropriate.

2. The ISS-LIS data used in this study are non-quality controlled. That could, to some degree, affect the results. I would suggest adding the reference of Blakeslee's talk at the 2019 GLM meeting (https://goes-r.nsstc.nasa.gov/home/meetingagenda-2019), which showed that ISS-LIS is slightly less sensitive than TRMM, and have a few sentences discussing the possible impact.

=> Although 'officially' named as "non-quality controlled" data, it is somewhat misleading. The data does include some quality control in the form of alert/warning flags, however some improvements to the algorithm creating the ISS-LIS data files are still ongoing. The main issue being the sensitivity of ISS-LIS compared to TRMM-LIS, or as formulated by Blakeslee in his 2019 GLM presentation: "The DE for LIS on ISS is on the order of 5% less than the DE for TRMM LIS, reflecting its slightly lower sensitivity. We expect that difference to decrease some when we apply our refined view time corrections to the data associated with the solar panel obscuration presently not being accounted for in view time". Note that the final quality controlled data are currently unavailable, but will be made available at a later undefined date. The following is added to the text at L127: "The non-quality controlled label is somewhat misleading since the data does include quality information in the form of alert/warning flags, however some improvements to the algorithm creating the ISS-LIS data files is still ongoing. The main issue being the sensitivity of ISS-LIS compared to TRMM-LIS, with the DE for LIS on ISS on the order of 5% less than the DE for TRMM LIS (R. Blakeslee, 2019 GLM Annual Science Team Meeting, Huntsville), impacting to some degree the DE values presented in this paper."

3. In Line 206, it says that "on average an ISS-LIS group occurs first." This looks a little odd to me. LIS group is an accumulation of a 2 ms period and the LIS timestamp is adjusted to the middle of that frame, which is 1 ms earlier than the end of the frame (Bitzer and Christian, 2015). So my impression is that the radio signals should be detected earlier. Zhang et al. (2016) also shows similar results. Do you have an explanation why ISS-LIS group occurs first in here? Possible scenarios could be that the EUCLID didn't report the initial breakdown processes in the flashes, while LIS reported them. If that's the case, then what is the detection efficiency of EUCLID reporting initial breakdown that NLDN? I would suggest the authors double-check the results. If the results are correct, then this could lead to some further discussions.

=> Previously, the best match between a EUCLID stroke and an ISS-LIS group was taken to be the closest in time. However, we believe it should be the closest in space. Changing the 'best' choice of match, brings the outcome in line with the expectations whereas now a matched ISS-LIS group is detected slightly later than the corresponding EUCLID stroke. Fig. 5 has been changed accordingly, and the text has been adapted at L170 as follows: "Note that for the stroke/group DE comparisons one ISS-LIS group can be matched to several EUCLID strokes/pulses and vice versa but for time/Location accuracy comparisons only the closest discharge in space is used." & at L206: "time offset of 0.23 ms (0.11 ms). Thus, on average a EUCLID stroke occurs first."

Minor comments:

1. Line 41: Brightness is fine, but the LIS radiance is actually energy density (Koshak, 2010).
=> The sentence now reads as "The spectral energy density for each event is also available (Koshak et al., 2010), ...".

**2. Line 216: artifact**

=> Artefact is British, whereas artifact is American spelling. AMT is a European journal, so we remain using artefact in the text.

3. Line 228: Should it be 12.9% and 20.5%, respectively? The P(EUCLID) in the full domain should be less than in the centre of the network, correct?

=> Correct, it should have been 12.9% and 20.5%, respectively. However, after some additional consideration and private communication with other scientists after the initial manuscript submission, our current understanding is that the absolute DE values at the level of groups/strokes are not relevant and inappropriate to use. This is due to the fact that the number of groups per LLS report is very large; artificially increasing the ISS-LIS group abs DE value. Therefore we have opted to remove the abs DE values for strokes/groups in Section 4.1 in the reviewed manuscript. The results for the relative DE remain in the text.

4. The very first paragraph on Page 9 (line number around 255) that discussed the absolute DE of ISS-LIS. The fact that ISS-LIS is less sensitive than TRMM-LIS might also contribute to the findings here. I would suggest adding that in the paragraph.

=> The text has been changed slightly into: "Additionally, note that in Zhang et al. (2016), the NLDN observations used were restricted to the areas where the NLDN detection efficiency is highest. Hence, P(ISS-LIS) and P(EUCLID) of 63.4% and 69.4%, respectively, in this work should be compared to P(TRMM-LIS) and P(NLDN) of 81.5% and 58.2%, respectively, in Zhang et al. (2016). Taking into account the fact that ISS-LIS is somewhat less sensitive compared to TRMM-LIS, would bring the latter values a bit more in line with each other."

---

## Author Comment (AC2) · 30 Mar 2020

**REVIEWER 2**

• Introduction (lines 67-69): in my opinion, the authors should provide more details about Erdmann et al. (2019) work, in which data from ISS-LIS were compared against observations from on-ground lightning detection networks. The authors should better emphasize the "added-value" of their work compared to the previous study just mentioned.

**=> In the introduction it has been now mentioned that Erdmann's work (2019) was the first time that data from ISS-LIS and a ground-based LLS network were compared, albeit over a shorter time period and spatial area than our present study: "The data of this sensor, called ISS-LIS, was analyzed in Erdmann et al. (2019) for the time period March 2017 to March 2018. They compared for the first time, although in a small part of Europe, ISS-LIS data to the low-frequency LLS of Météorage and the lightning mapping array SAETTA (Coquillat et al., 2019) over the northwestern Mediterranean Sea near Corsica. In this paper the performance of EUCLID (EUropean Cooperation for LIghtning Detection), a ground based LLS similar to the LLS of Météorage, will be evaluated relative to the ISS-LIS data over an extended time period and larger area in Europe compared to Erdmann et al. (2019)." Relative flash DE values as described in Erdmann et al. (2019) were mentioned already throughout the text at appropriate places, but we have added now the different time & spatial matching criteria in the text as well: "Applying analogue flash matching criteria as in Erdmann et al. (2019) over the Corsica area, i.e., strokes/pulses are matched to ISS-LIS groups using a 1.0 s temporal and 20 km spatial criterion,"**

• Data (2.1 Euclid): Why did the authors choose the data from EUCLID network for their analysis? I suggest to provide clear and strong motivations about this choice. It is well known that in European area other lightning detection networks are available, providing data about cloud-to-ground (CG) and intra-cloud (IC) flashes with high detection efficiency.

**=> There are definitely other ground-based LLS covering Europe. However, the performance of EUCLID is monitored frequently through various ground-truth campaigns (either via tower measurements or via mobile E-field and video measurements) over many years and is to our knowledge, the best documented network to date in Europe in terms of its location accuracy and detection efficiency, as published in various journal articles and conference proceedings. The results of those ground-truth campaigns were mentioned already in the manuscript and is believed to be sufficient why EUCLID has been chosen.**

• Data (2.2 LIS, Lines 124): the authors stated that they used a non-quality controlled ISS-LIS dataset. I think that some clarifications are needed. What does mean "non-quality controlled"? How did the authors overcome this problem?

**=> Although 'officially' named as "non-quality controlled" data, it is somewhat misleading. The data does include some quality control in the form of alert/warning flags, however some improvements to the algorithm creating the ISS-LIS data files are still ongoing. The main issue being the sensitivity of ISS-LIS compared to TRMM-LIS, or as formulated by Blakeslee in his 2019 GLM presentation: "The DE for LIS on ISS is on the order of 5% less than the DE for TRMM LIS, reflecting its slightly lower sensitivity. We expect that difference to decrease some when we apply our refined view time corrections to the data associated with the solar panel obscuration presently not being accounted for in view time". Note that the final quality controlled data are currently unavailable, but will be made available at a later undefined date. The following is added to the text at L127: "The non-quality controlled label is somewhat misleading since the data does include quality information in the form of alert/warning flags, however some improvements to the algorithm creating the ISS-LIS data files is still ongoing. The main issue being the sensitivity of ISS-LIS compared to TRMM-LIS, with the DE for LIS on ISS on the order of 5% less than the DE for TRMM LIS (R. Blakeslee, 2019 GLM Annual Science Team Meeting, Huntsville), impacting to some degree the DE values presented in this paper."**

• Methodology (Line 168): I suggest to produce a Figure or a Table to support and justify these choices about spatial and temporal criteria.

**=> The temporal and spatial criteria are similar to the once used in previously published journal papers as cited in the text, i.e., Franklin (2013), Bitzer et al. (2016), Zhang et al. (2016, 2019).**

• Results: to improve the quality of the presentation of the findings of this study, I propose to produce one or two additional tables.
**=> Two extra tables (1 and 2) including the stroke/group and flash DE values are added to the text.**

• Conclusions: please add a brief discussion about the future implications of this work. I think it may have a good impact from different perspectives. Therefore, the conclusions section should not be limited to a summary of the main results.
**=> At the end of Section 5 following is added: "The main objective of this study was to investigate how well the observations of a ground-based LLS, in this case EUCLID, are linked to space-based optical lightning signatures of LIS on ISS over a large part in western Europe. Consequently, the method described in this work and results thereof provide a framework to be used in potential future studies involving MTG-LI observations over Europe. Moreover, whereas now the focus was on one particular LLS, it is straightforward to apply it to observations of different ground-based LLSs covering the entire MTG-LI domain or parts thereof."**

• Finally, I suggest to carefully check the paper to address some minor typos
**=> Editorial changes are addressed.**